# Prediction of Hemorrhagic Transformation after Ischemic Stroke Using Machine Learning

**DOI:** 10.3390/jpm11090863

**Published:** 2021-08-30

**Authors:** Jeong-Myeong Choi, Soo-Young Seo, Pum-Jun Kim, Yu-Seop Kim, Sang-Hwa Lee, Jong-Hee Sohn, Dong-Kyu Kim, Jae-Jun Lee, Chulho Kim

**Affiliations:** 1Department of Convergence Software, Hallym University, Chuncheon 24252, Korea; jeong5905@gmail.com (J.-M.C.); syseo96@gmail.com (S.-Y.S.); yskim.hallym@gmail.com (Y.-S.K.); 2Institute of New Frontier Research Team, Hallym University College of Medicine, Chuncheon 24252, Korea; pumjun4093@gmail.com (P.-J.K.); bleulsh@naver.com (S.-H.L.); deepfoci@hallym.or.kr (J.-H.S.); doctordk@naver.com (D.-K.K.); iloveu59@hallym.or.kr (J.-J.L.); 3Department of Neurology, Chuncheon Sacred Heart Hospital, Chuncheon 24253, Korea; 4Department of Otorhinolaryngology and Head and Neck Surgery, Chuncheon Sacred Heart Hospital, Chuncheon 24253, Korea; 5Department of Anesthesiology and Pain Medicine, Chuncheon Sacred Heart Hospital, Chuncheon 24253, Korea

**Keywords:** stroke, hemorrhagic transformation, machine learning, deep learning, neural network

## Abstract

Hemorrhagic transformation (HT) is one of the leading causes of a poor prognostic marker after acute ischemic stroke (AIS). We compared the performances of the several machine learning (ML) algorithms to predict HT after AIS using only structured data. A total of 2028 patients with AIS, who were admitted within seven days of symptoms onset, were included in this analysis. HT was defined based on the criteria of the European Co-operative Acute Stroke Study-II trial. The whole dataset was randomly divided into a training and a test dataset with a 7:3 ratio. Binary logistic regression, support vector machine, extreme gradient boosting, and artificial neural network (ANN) algorithms were used to assess the performance of predicting the HT occurrence after AIS. Five-fold cross validation and a grid search technique were used to optimize the hyperparameters of each ML model, which had its performance measured by the area under the receiver operating characteristic (AUROC) curve. Among the included AIS patients, the mean age and number of male subjects were 69.6 years and 1183 (58.3%), respectively. HT was observed in 318 subjects (15.7%). There were no significant differences in corresponding variables between the training and test dataset. Among all the ML algorithms, the ANN algorithm showed the best performance in terms of predicting the occurrence of HT in our dataset (0.844). Feature scaling including standardization and normalization, and the resampling strategy showed no additional improvement of the ANN’s performance. The ANN-based prediction of HT after AIS showed better performance than the conventional ML algorithms. Deep learning may be used to predict important outcomes for structured data-based prediction.

## 1. Introduction

According to the Global Burden of Stroke in the World Health Organization’s 2016 report, stroke is the leading cause of death and disability worldwide [1], with the incidence of ischemic stroke exceeding that of hemorrhagic stroke [2]. Hemorrhagic transformation (HT) is the one of the major potential complications after acute ischemic stroke (AIS), and is associated with the natural recanalization of the occluded cerebral arteries, thrombolysis, or mechanical thrombectomy, and is a major barrier for antithrombotic treatment after AIS [3,4,5]. Therefore, it is an important issue for stroke practitioners to predict the occurrence of HT during treatment in these patients [6,7]. However, in previous studies, the performance of predicting HT via C-statistics showed relatively poor predictive power at 0.70 [8].

Recently, machine learning (ML) or deep learning (DL) algorithms have been widely used in medical practice as a clinical decision support system [9,10]. In several studies, the usefulness of the ML strategy to predict the risk of HT following AIS was assessed [11,12,13,14,15]. Wang et al. reported that the neural network model showed the best performance (AUROC = 0.82) to predict symptomatic intracerebral hemorrhage (ICH) following thrombolysis in patients with AIS [11]. In another multicenter trial using the Observational Medical Outcomes Partnership Common Data model, the least absolute shrinkage and selection operator regression model showed an AUROC of 0.75 to predict HT [12]. Asadi et al. studied the usefulness of ML algorithms to predict poor outcomes in patients with AIS who received endovascular intervention [13]. They suggested that the support vector machine (SVM) successfully predicted poor outcomes, and that post-infarct ICH was an important factor in a poor prognosis. However, this study had a relatively small number of study participants (107 subjects) and, thus, the study result may require additional validation. Other studies reported a high accuracy rate (~84%) for predicting HT in their stroke cohort, with only the radiologic markers of an MRI used to perform the ML tasks [14,15]. In this regard, different ML algorithms were used to improve the prediction of HT after AIS. However, there are no studies showing high prediction performance using clinical variables in ML tasks.

HT can be divided into symptomatic and asymptomatic cases. Previous studies reported that not only symptomatic HT, but also asymptomatic HT can affect clinical outcomes after AIS [7,16]. There are cases where intracranial hemorrhage, occurring after cerebral reperfusion, could be asymptomatic [17], and, in these cases, it is difficult to determine when to begin antithrombotic treatment [18]. Therefore, we limited HT to a radiological definition rather than a clinical definition. We hypothesized that DL algorithms could better predict HT after AIS than conventional prediction models. Thus, we aimed to assess the important predictor of HT in several ML algorithms, and how to improve the prediction performance of the ML model used in this study.

## 2. Materials and Methods

### 2.1. Population and Study Design

This study is a cross-sectional retrospective case-control study using a prospectively collected stroke database in a tertiary teaching hospital. In this registry, patient’s demographics, stroke mechanism, clinical, laboratory, and radiological results were collected by the stroke practitioner and regularly audited by external researchers [19]. From January 2015 to December 2020, a total of 2555 patients admitted to this hospital were included in the registry. Among them, patients with diffusion restrictive lesions in brain MRI scans, with relevant focal neurologic deficits, were included in the analysis. In this analysis, we excluded patients admitted to the hospital seven days after stroke onset and those with missing variables in clinical and laboratory parameters (Figure 1). This study was approved by the Chuncheon Sacred Heart Hospital Institutional Review Board/Ethics Committee (IRB No. 2019-11-017). Written informed consent for the registry enrollment was provided by the participants or their guardians.

### 2.2. Data Information

Clinicodemographic variables, including age, sex, body mass index, and cardiovascular disease risk profile at hospital admission, were included in the ML model. Age, sex, and stroke-related information, including a history of taking antithrombotics, symptom onset to hospital arrival time, stroke subtype according to the Trial of ORG 10172 in Acute Stroke Treatment classification, and laboratory parameters at hospital admission were also included in the ML model. Originally, HT was divided into hemorrhagic infarct type 1, 2, parenchymal hemorrhage type 1, 2, and symptomatic ICH according to the second European Co-operative Acute Stroke Study-II criteria [20]. We defined HT as when all of these subtypes of HT were identified in follow up brain CT or gradient-echo MRI 48 h after the initial evaluation of AIS.

### 2.3. Machine Learning Algorithm

As described earlier, we aimed to assess the classification performance (HT or no HT) of several ML algorithms using different optimization techniques. At first, we randomly divided a whole dataset into a training and test dataset with a 7:3 ratio, with a similar proportion of HT maintained in the training and test dataset. For the input variables, continuous variables were used as the raw values, and categorical variables were encoded using one-hot encoding. In the preprocessing process of the continuous variables, we used these as raw values (crude method) for the different scaling methods, including normalization, min-max scaling, standardization, and robust scaling (Figure 2a) [21,22]. Binary logistic regression (BLR), SVM, extreme gradient boosting (XGB), and artificial neural network (ANN) algorithms were used to assess the performance of each algorithm in predicting HT in our dataset. The ANN algorithm was composed of an input layer, four fully connected hidden connected layers, and one output layer (Figure 2b) in the ANN preprocessing task. In the training process, we used five-fold cross validation to reduce the model’s overfitting, and used the grid search technique to select the best combination of hyperparameters in each ML algorithm. The detailed information on the parameter settings are presented in Appendix A. On performing each ML task, we extracted the variable importance of the input variables to identify which variables were important in predicting HT in the training dataset. We used the sklearn and keras Python package for these ML processes, and the model training was performed with the TensorFlow interface using NVIDIA’s GeForce GTX 1080ti graphic processing units.

### 2.4. Statistical Analysis

The baseline characteristics of the patients in the training and test datasets were compared using the Student’s t-test or the Mann–Whitney U test for continuous variables, and Pearson’s χ^2^-test for categorical variables, as appropriate. When we obtained the probability for the HT from each ML classifier, values of >0.5 were assigned positive HT status. The performance of each ML model was measured with the receiver operating characteristics curve. All statistical analyses were performed with R version 3.6.1 (the R Foundation for Statistical Computing) and Python version 3.7.7 in the anaconda environment.

## 3. Results

A total of 2028 patients were included in the final ML tasks. Age and portion of male were 69.6 years and 58.3%, respectively. In the whole dataset, HT was observed in 318 patients (15.7%). The comparison of baseline characteristics between the training and test datasets are presented in Table 1. Stroke subtype and stroke severity were equally distributed in the training and test datasets. In addition, the proportion of patients who had been taking antithrombotics before the index stroke or who received thrombolysis for the index stroke were also equally distributed between the training and test datasets. Therefore, there was no significant difference in the input variables for the HT prediction model.

Table 2 shows the overall performance of HT prediction for each of the ML classifiers. The performance of the grid search-based ANN algorithm was the best classifier for predicting HT (accuracy = 87.8%, F1-score = 93.2%), followed by the SVM algorithm. In addition, we represented the most important variables in each of the ML classifiers (Table 3). Although the variable importance factors in each ML algorithm were different, gender, age, prior antithrombotic usage, stroke severity, white blood cell count, stroke subtype, and fasting blood sugar were identified as important factors in the model’s classification.

Figure 3 shows the result of the performance of each ML classifier with a five-fold cross validation and grid search hyperparameter optimization technique. The ANN algorithm was the best performing algorithm on the test dataset (AUROC = 0.842, Figure 3a,b). We additionally performed ANN modelling with scaling of the input variables, and there was no additional improvement in the model’s performance (Figure 3b). We performed ML tasks to determine whether scaling of the input parameters or resampling technique could improve ML algorithm performances. The implementation of these techniques in our BLR, SVM, and XGB algorithms did not show any additional improvement in the model’s performances (Appendix A).

## 4. Discussion

In this study of the performance comparison of HT prediction in AIS patients, we identified that the ANN’s prediction performance was better than those of other ML algorithms. In addition, the grid-search hyperparameter optimization technique was useful for improving the performance of ML algorithms using structural numerical data, but the scaling strategy did not show any additional improvement.

It is important for AIS patients who have completed the emergent treatment to reduce the incidence of complications, such as pneumonia, deep vein thrombosis, or HT, which is known to be associated with the worsening of the stroke prognosis. The incidence of HT ranges from 11.0 to 37.5 in patients with AIS according to different clinical settings [23,24,25,26,27,28]. Antithrombotic therapy to prevent additional ischemia immediately after the index stroke is associated with the development of HT or intracranial hemorrhage. On the other hand, the risk of stroke recurrence is high during the acute stage and more than half of relapsed patients have a recurrence within 30 days of the index stroke [29]. Therefore, to minimize the impact of HT after a stroke, we should consider which variables have a causal relationship with the HT development using the conventional statistical model, and which ML models are effective at improving the prediction performance of subsequent HT development.

The ANN algorithm had several advantages compared to traditional ML algorithms. First, the ANN algorithm is quite robust to noise in the training data [30]. If the training data contains errors, they do not significantly affect the final result of the algorithm. Second, ANN is resilient to long duration training processes due to the considerable number of parameter weights and training examples [31]. Third, the higher the number of hidden layers stacked in the ANN algorithm, the more chance that vanishing gradient problems could develop. However, ANN algorithms with few hidden layers, utilizing structured numerical data, can overcome these problems [32]. There is no exact information about how many layers can be stacked to overcome the disadvantage of ANN algorithms falling into the local minima. However, in the case of using medical structured data, the performance of ANN algorithms is reported to be superior when using 3–5 hidden layers, as in this study [33,34]. In addition, the ANN algorithm can perform complex non-linear fitting of high dimensional data, and has well-developed architecture selection methods to prevent the overfitting of training models [35].

Wang et al. studied the usefulness of ML algorithms to predict symptomatic ICH after thrombolysis in 2237 patients with hyperacute ischemic stroke [11]. Of these ML models, the three-layered ANN model showed the best performance in terms of predicting symptomatic ICH in this cohort. This study was conducted on a different stroke population and, thus, a direct comparison of the algorithm’s performance may be difficult. However, these difference in the ANN model’s performance could be explained by the following reasons. First, Wang et al.’s study used imputation for missing parameters. Since missing values were replaced with representative values, such as mean or median values during multiple imputation, the meanings of the variables may have flattened during this imputation process and, as a result, the performance of the ML classifier could be underestimated [36]. Second, the absence of stroke-related information, such as stroke subtype or laboratory parameters, in Wang et al.’s study might be associated with lower resolution for HT prediction. Indeed, stroke subtype classified with the Trial of Org 10172 in Acute Stroke Treatment classification is associated with the occurrence of HT, with cardioembolic stroke etiology having a causal relationship with the HT development [37]. Third, we used five-fold cross validation and the grid search technique for hyperparameter optimization, which enabled us to obtain tuned performance for each ML algorithm by reducing the overfitting of training data.

In general, scaling methods, such as standardization or normalization, reduce the variability of the weight or error of each variable, thereby reducing the failure of the learning process due to gradient exploding that occurs during learning of the neural network model and improving the performance of the model [38]. However, the learning result of additional ANN models performed with the scaling input variable in our study were not better than the crude ANN model. In other DL studies relating to stroke, there was no mention of the effect of neural network scaling on DL performance [11,39]. Ahsan et al. reported the effect of scaling on performance in various ML methods [40]. They concluded that the effect of scaling on ML performance varied depending on the characteristics of the data. Therefore, it can be reported as evidence that the scaling method of ANN had no effect on the model performance improvement of the numerical data of stroke patients.

Of our ML models, age, gender, stroke severity, stroke subtype, prior antithrombotics usage, white blood cell count, and fasting blood sugar were important variables in predicting HT (Table 3). Age and stroke severity are important prognostic markers for AIS [41], and were also identified as a predictor for HT in other studies [42]. Andrade et al. summarized the important predictors for HT in clinical trials [43]. Considering that important variables in our ML model are exactly matched with the variables presented in this report, we suggest that clinically important variables have a significant influence on the performance of the classifier, even in ML classification.

There are several limitations to our study. First, we only evaluated the numerical or categorical data related to the patients’ clinicodemographic factors for laboratory variables at admission. HT can be affected by a variety of post-stroke management treatments, such as blood pressure management and concurrent post-stroke antithrombotic medications [44,45]. Therefore, we could not evaluate the impact of post-stroke care for the development of HT. Second, we did not assess radiologic markers of HT. Aside from our research on HT, image DLs using CT or MRI are being actively conducted. In future studies, we expect that the ensemble learning method, which adds the patient’s clinical variables and image variables, will further enhance the predictive power of the DL model.

## 5. Conclusions

The ANN algorithm was more effective at predicting HT in AIS patient then the conventional ML algorithms and showed the best performance for the prediction of HT in our dataset (0.844) without additional feature scaling. In later trials, ensemble strategy, using numerical and unstructured imaging data DL, could be useful to predict HT after AIS.

## Figures and Tables

**Figure 1 jpm-11-00863-f001:**
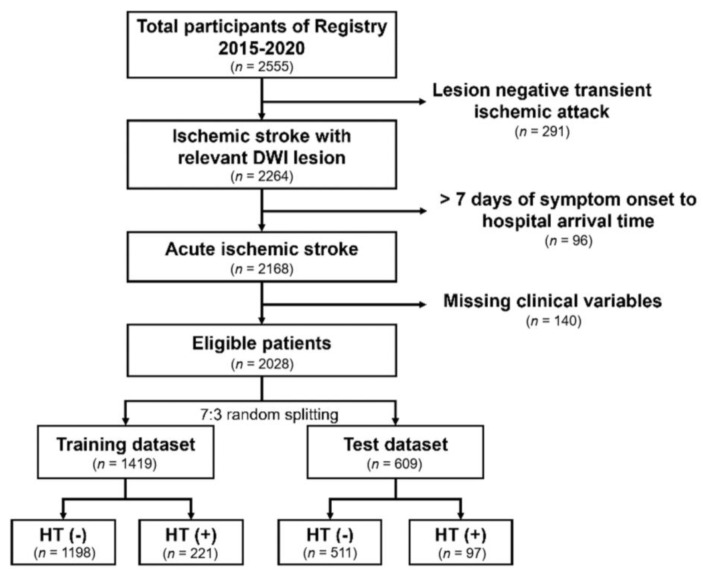
Flowchart of the study participants. DWI: diffusion weighted image; HT: hemorrhagic transformation.

**Figure 2 jpm-11-00863-f002:**
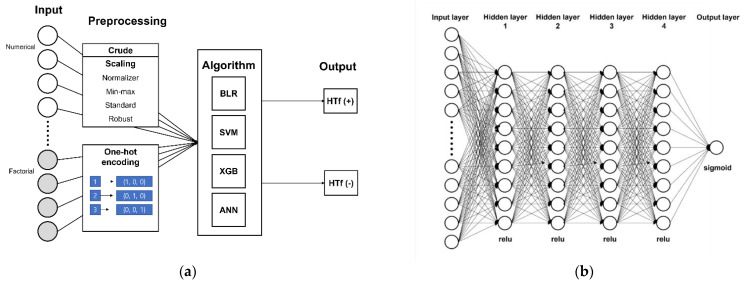
Schematic representation of the machine learning model: (**a**) the preprocessing process of the categorical and continuous variables and (**b**) the schematic representation of the artificial neural network structure. BLR: binary logistic regression; SVM: support vector machine; XGB: extreme gradient boosting; ANN: artificial neural network; HTf: hemorrhagic transformation; and relu: rectified linear unit.

**Figure 3 jpm-11-00863-f003:**
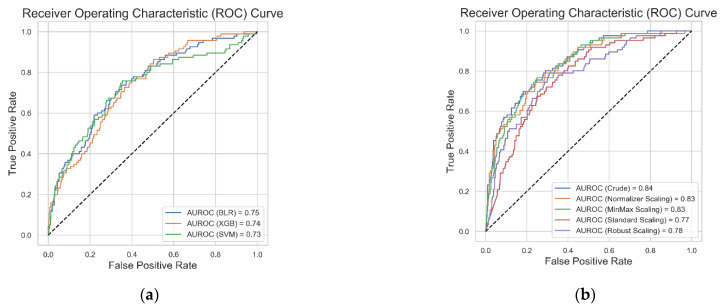
Result of the receiver operating characteristic curve of binary logistic regression, extreme gradient boosting, and support vector machine algorithms (**a**) and the artificial neural network algorithm before and after input parameter scaling (**b**). BLR: binary logistic regression; XGB: extreme gradient boosting; SVM: support vector machine; ANN: artificial neural network; and AUROC: area under the receiver operating characteristic curve.

**Table 1 jpm-11-00863-t001:** Comparison of baseline characteristics between training and test datasets.

Variables	Training (*n* = 1419)	Test (*n* = 609)	Whole Dataset (*n* = 2028)	*p* Value
Male	815 (57.4%)	368 (60.4%)	1183 (58.3%)	0.229
Age, year	69.7 ± 12.9	69.3 ± 12.4	69.6 ± 12.8	0.451
Onset to arrival time, hours	29.1 ± 44.5	32.2 ± 45.8	30.6 ± 48.2	0.183
BMI, kg/m^2^	24.1 ± 3.6	24.1 ± 3.4	24.1 ± 3.6	0.606
Initial NIHSS, score	5.1 ± 5.7	4.9 ± 5.6	5.0 ± 5.6	0.562
Stroke subtype				0.313
LAA	491 (34.6%)	222 (36.5%)	713 (35.2%)	
SVO	410 (28.9%)	185 (30.4%)	595 (29.3%)	
CE	270 (19.0%)	111 (18.2%)	381 (18.8%)	
SOE	51 (3.6%)	12 (2.0%)	63 (3.1%)	
SUE	197 (13.9%)	79 (13.0%)	276 (13.6%)	
Past medical history				
Prior stroke	359 (25.3%)	146 (24.0%)	505 (24.9%)	0.564
Hypertension	921 (64.9%)	398 (65.4%)	1319 (65.0%)	0.834
Diabetes	250 (17.6%)	118 (18.3%)	368 (18.1%)	0.167
Dyslipidemia	495 (34.9%)	208 (34.2%)	703 (34.7%)	0.979
Current smoking	319 (22.5%)	140 (23.0%)	459 (22.6%)	0.847
Atrial fibrillation	273 (19.2%)	105 (17.2%)	378 (18.6)	0.319
Prior antithrombotics treatment	529 (37.3%)	222 (36.5%)	751 (37.0%)	0.762
Thrombolysis	188 (13.2%)	76 (12.5%)	264 (13.0%)	0.689
Laboratory parameter				
WBC, 10^3^/μL	7.8 ± 2.9	7.9 ± 3.0	7.8 ± 2.9	0.414
Hemoglobin, g/dL	13.6 ± 2.0	13.8 ± 1.8	13.7 ± 2.0	0.140
Platelet count, 10^3^/μL	233.6 ± 74.9	234.5 ± 80.9	233.9 ± 76.8	0.820
Total cholesterol, g/dL	168.1 ± 63.7	168.2 ± 41.5	168.2 ± 57.9	0.994
TG, mg/dL	128.8 ± 85.5	133.1 ± 81.3	130.1 ± 84.3	0.288
HDL, mg/dL	45.7 ± 11.5	44.9 ± 10.6	45.5 ± 11.3	0.158
LDL, mg/dL	100.3 ± 35.4	102.4 ± 34.9	100.9 ± 35.2	0.225
BUN, mg/dL	17.7 ± 9.4	17.6 ± 9.3	17.7 ± 9.4	0.860
Creatinine, mg/dL	1.0 ± 0.8	1.0 ± 0.7	1.0 ± 0.7	0.956
FBS, mg/dL	126.7 ± 52.8	126.0 ± 49.0	126.5 ± 51.6	0.759
A1c, %	6.3 ± 1.4	6.3 ± 1.4	6.3 ± 1.4	0.848
INR	1.1 ± 0.4	1.0 ± 0.2	1.1 ± 0.3	0.235
SBP, mmHg	146.0 ± 26.5	145.6 ± 26.4	145.9 ± 26.5	0.768
DBP, mmHg	84.0 ± 13.9	83.9 ± 14.1	84.0 ± 13.9	0.522
Hemorrhagic transformation	221 (15.6%)	97 (15.9%)	318 (15.7%)	0.893

Categorical variables are represented by the number (percent), and continuous variable are represented by mean (± standard deviation). BMI: body mass index; NIHSS: National Institute of Health Stroke Scale; LAA: large artery atherosclerosis, SVO: small vessel occlusion; CE: cardioembolism; SOE: stroke of other determined etiology; SUO: stroke of undetermined etiology; WBC: white blood cell; TG: triglycerides; HDL: high-density lipoprotein; LDL: low-density lipoprotein; BUN: blood urea nitrogen; FBS: fasting blood sugar; A1c: glycated hemoglobin; INR: international normalized ratio; SBP: systolic blood pressure; and DBP: diastolic blood pressure.

**Table 2 jpm-11-00863-t002:** Results of several performance parameters of machine learning algorithms to predict hemorrhagic transformation in the test dataset.

	TP	FP	FN	TN	Total	Precision	Recall	Accuracy	F1-Score
BLR	486	28	71	24	609	87.3	94.6	83.7	90.8
SVM	504	10	78	17	609	86.6	98.1	85.6	92.0
XGB	486	28	73	22	609	86.9	94.6	83.4	90.6
ANN_crude	506	17	57	29	609	89.9	96.7	87.8	93.2

TP: true positive; FP: false positive; FN: false negative; TN: true negative; BLR: binary logistic regression; SVM: support vector machine; XGB: extreme gradient boosting; and ANN crude: artificial neural network crude model.

**Table 3 jpm-11-00863-t003:** Most important input variables of predicting hemorrhagic transformation after acute ischemic stroke in each machine learning classifier.

No	Variable	BLR	SVM	XGB	ANN
1	Age	3rd	7th	1st	
2	Male	1st	5th	8th	
3	Onset to arrival time				
4	BMI				
5	NIHSS		1st	3rd	1st
6	Previous mRS	7th			
7	TOAST_1				
8	TOAST_2				2nd
9	TOAST_3		2nd		5th
10	TOAST_4	8th	9th	2nd	
11	TOAST_5				8th
12	Previous stroke	10th			
13	Hypertension				
14	Diabetes	4th			
15	Dyslipidemia		6th	9th	
16	Current smoking				
17	Atrial fibrillation				7th
18	Prior antithrombotic usage	2nd		4th	
19	Thrombolysis	9th	10th		
20	WBC		3rd		6th
21	Hemoglobin			5th	10th
22	Platelet count		8th		9th
23	Total cholesterol				
24	Triglycerides				
25	High density lipoprotein			6th	
26	Low density lipoprotein	5th			
27	Blood urea nitrogen		4th		
28	Creatinine				
29	Fasting blood sugar				3rd
30	Glycated hemoglobin			7th	
31	INR				
32	BPsys			10th	4th
33	BPdia	6th			

BLR: binary logistic regression; SVM: support vector machine; XGB: extreme gradient boosting; ANN: artificial neural network; BMI: body mass index; NIHSS: National Institute of Health Stroke Scale; mRS: modified Rankin Scale; TOAST: Trial of ORG 10172 in Acute Stroke Treatment; WBC: white blood cell; INR: international normalized ratio; BPsys: systolic blood pressure; and BPdia: diastolic blood pressure.

## Data Availability

The data presented in this study are available on request from the corresponding author. The data are not publicly available due to the policy of our IRB.

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
