# Peer review of "Prediction of Hemorrhagic Transformation after Ischemic Stroke Using Machine Learning"

_jpm, 2021, doi:10.3390/jpm11090863_

Round 1
Reviewer 1 Report
The authors present an interesting work about an important topic using a new technology focus. There are some issues that I think it should be solved:
- Introduction: Large clinical trials didn't show a significant difference between thrombectomy and BMT in symptomatic hemorrhage.
- Methods: Why authors selected hemorrhagic transformation as outcome instead of sympomatic HT. As a stroke neurologist, sHT is the main concern when we treat a patient or start antithrombotic therapy. Mixing hemorrhagic infarction (HI-1, maybe related with better outcome) with parenchymal hematoma with mass effect (PH2, high rate of mortality) reduces the clinical impact of the findings.
- Discussion: Mixing again studies about HT and symptomatic HT is not apropiate and it should be refocused in studies with the same outcome.
Author Response
The authors present an interesting work about an important topic using a new technology focus. There are some issues that I think it should be solved:
- Introduction: Large clinical trials didn't show a significant difference between thrombectomy and BMT in symptomatic hemorrhage.
Methods: Why authors selected hemorrhagic transformation as outcome instead of symptomatic HT (sHT). As a stroke neurologist, sHT is the main concern when we treat a patient or start antithrombotic therapy. Mixing hemorrhagic infarction (HI-1, maybe related with better outcome) with parenchymal hematoma with mass effect (PH2, high rate of mortality) reduces the clinical impact of the findings.
- R) We appreciate your thoughtful comments. At first, we also focused on the prediction of sHT after AIS using ML algorithms. However, there were several reports that asymptomatic HT may be related to the clinical outcomes. In addition, asymptomatic ICH can be observed after cerebral reperfusion. In this regard, we limited HT to a radiological definition rather than a clinical definition because excluding asymptomatic HT(or ICH) could reduce the homogeneity of HT.
To meet your suggestion, we added some sentences for this.
“Recently, machine learning (ML) or deep learning (DL) algorithms are widely used in medical practice as the name of clinical decision support system.8, 9 In several studies, usefulness of ML strategy to predict the risk of HT following AIS.10-14 Wang et al. reported that neural network model showed the best performance (area under the receiver operating characteristic; AUROC=0.82) to predict symptomatic intracerebral hemorrhage following thrombolysis in patients with AIS.10 In another multicenter trial using Observational Medical Outcomes Partnership Common Data Model, least absolute shrinkage and selection operator regression model showed the AUROC of 0.75 to predict HT.11 In this regard, different ML algorithms have been used to improve the prediction of HT after AIS. HT can be divided into symptomatic and asymptomatic cases. Previous studies have reported that not only symptomatic HT, but also asymptomatic HT can affect clinical outcome after AIS.7, 15 Besides, There are cases where intracranial hemorrhage that occurs after cerebral reperfusion could be asymptomatic,16 and it is difficult to decide when to start antithrombotic treatment in this case.17 Therefore, we limited HT to a radiological definition rather than a clinical definition. We hypothesized that DL algorithms could be better to predict HT after AIS than conventional prediction model. Thus, we aim to assess the important predictor of HT in several ML algorithms, and how to improve the prediction performance of the ML model in this study.”
- Discussion: Mixing again studies about HT and symptomatic HT is not appropriate and it should be refocused in studies with the same outcome.
- R) Thank you for your thoughtful comment. Indeed, we identified that the research results on sHT and HTs are mixed and described in the discussion section of our manuscript. Therefore, we deleted a few sentences to mainly describe studies on HT as the reviewer’s suggestion. However, there are few studies predicting HT using ML, so Wang's report was remained and compared the performance of the ML classifier, not focused on the predictions of HT or sICH.
“Wang et al. studied the usefulness of ML algorithms to predict symptomatic ICH after thrombolysis in 2237 patients with hyperacute ischemic stroke.10 Of these ML models, three-layered ANN model’s performance showed the best performance to predict symptomatic ICH in these cohort. Those study was conducted on different stroke population, and direct comparison of the algorithm’s performance may be difficult. However, these difference in ANN’ model’s performance could be explained by the following reasons. At first, Wang’s study used imputation for missing parameters. Because missing values are replaced with representative values such as mean or median values during multiple imputation, the meanings of variables may be flattened during this imputation process, and as a result, the performance of ML classifier can be underestimated.33 Second, absence of stroke-related information such as stroke subtype or laboratory parameters in Wang’s study might be associated with lower resolution for HT prediction. Indeed, stroke subtype classified with the Trial of Org 10172 in Acute Stroke Treatment classification is associated with the occurrence of HT, of which cardioembolic stroke etiology has the causal relationship with the HT development.34 In third, we used 5-fold cross validation and grid search for hyperparameter optimization, which enables to obtain tunned performance for each ML algorithm by reducing the algorithms overfitting of training data. Chung et al. reported an AUROC of 0.91-0.94 to predict post-thrombolysis ICH and death in 331 patients using ANN algorithm.35 In this study, they used 5-fold cross validation, in which whole dataset was divided into 4 training folds and 1 validation fold. In this regard, as in those of Chung et al., the performance could be improved through cross validation and grid search technique.”
Reviewer 2 Report
Dear editor:
Thank you for inviting me to evaluate this article titled “Prediction of Hemorrhagic Transformation after Ischemic Stroke Using Machine Learning”. In this study, the authors proposed multiple machine learning algorithms to predict the risk of hemorrhagic transformation (HT) after acute ischemic stroke (AIS). However, as a machine learning study, the text is not well organized and the details of the model designs and results are not well presented. I think multiple issues need to be addressed before they can be accepted.
Major comments:
- In both Results and Materials & Methods sections, the wording of the machine learning algorithms is ambiguous. This is displayed in a few aspects:
- [line 97] “… Continuous variables were used as the raw values, and categorical variables were encoded using one-hot encoding in the ANN preprocessing task.” The authors described which parameters they used in their classifiers, but how they integrated/combined these parameters into the input feature is missing. It would be helpful if they could use a figure to provide the input information for their models.
- [line 100] “… and used grid search technique to select the best combination of hyperparameters in each ML algorithm”. It’s necessary to list the ranges of all hyperparameters used in the grid search for each ML algorithm.
- In figure 2B, the definitions of “Crude”, “Normalizer Scaling”, “MinMax Scaling”, “Standart Scaling” and “Robust Scaling” are missing.
- Since their input features include continuous variables as well as categorical variables. How did they scale these different types of variables as input features?
- In Figure 2, why did the authors not implement “Crude”, “Normalizer Scaling”, “MinMax Scaling”, “Standart Scaling” and “Robust Scaling” in BLR, SVM and XGB? Using these scalings of the input variables for all classifiers will make their conclusions more convincing.
- For a machine learning study, resampling the training and test dataset is essential to reduce sampling bias. A test comparing the four classifiers under multiple repetitions will consolidate their conclusions.
- The Introduction should be improved:
- The two paragraphs in the Introduction are not compact. I think the authors need to present the limitations and challenges of predicting HT with additional references at the end of the first paragraph.
- The background information about HT and AIS is general. Without sufficient background introduction, it is not easy for readers to intuitively understand why the authors use the baseline characteristics in Table 1.
- In the second paragraph, the authors need to discuss the shortcomings or limitations of the previous similar studies (such as references [10-14]), and point out the superiority of their model.
- Table S1 is interesting and important. It will be useful if the authors can further interpret the biological significance of these top variables in the ML classifiers.
Minor comments:
- In Table 1, the “BUN, mg/dL” of the training dataset equaling to “17.7 ± 941” might be a typo.
- The abbreviation (for example “AIS” in line 42) should be defined in the first occurrence.
Author Response
Thank you for inviting me to evaluate this article titled “Prediction of Hemorrhagic Transformation after Ischemic Stroke Using Machine Learning”. In this study, the authors proposed multiple machine learning algorithms to predict the risk of hemorrhagic transformation (HT) after acute ischemic stroke (AIS). However, as a machine learning study, the text is not well organized and the details of the model designs and results are not well presented. I think multiple issues need to be addressed before they can be accepted.
Major comments:
In both Results and Materials & Methods sections, the wording of the machine learning algorithms is ambiguous. This is displayed in a few aspects:
- [line 97] “… Continuous variables were used as the raw values, and categorical variables were encoded using one-hot encoding in the ANN preprocessing task.” The authors described which parameters they used in their classifiers, but how they integrated/combined these parameters into the input feature is missing. It would be helpful if they could use a figure to provide the input information for their models.
- R) We appreciate your comments on the summary for the preprocessing strategy of each ML algorithm. We also identified that the description of the pre-processing technique was not thoroughly described in our 1st manuscript. Therefore, we have added the corresponding figure as follows.
- [line 100] “… and used grid search technique to select the best combination of hyperparameters in each ML algorithm”. It’s necessary to list the ranges of all hyperparameters used in the grid search for each ML algorithm.
- R) Thank you for your comment, which is an important issue. We added parameters used in grid search technique in our ML classifiers as follows.
Table S2. Grid search parameters in each machine learning classifier.
|
|
BLR |
SVM |
XGB |
ANN |
|
C |
[0.001, 0.01, 0.1, 1, 10, 100] |
[0.001, 0.01, 0.1, 1, 10, 100] |
|
|
|
gamma |
|
[0.001, 0.01, 0.1, 1, 10, 100] |
|
|
|
kernel |
|
rbf, linear |
|
|
|
n_estimator |
|
|
range(5,30) |
|
|
max_depth |
|
|
range(6,10) |
|
|
learning rate |
|
|
[0.4, 0.45, 0.5, 0.55, 0.6] |
|
|
colsample_bytree |
|
|
[0.6, 0.7, 0.8, 0.9, 1.0] |
|
|
optimizer |
|
|
|
adam |
|
loss |
|
|
|
binary_crossentropy |
|
batch_size |
|
|
|
50 |
|
Epoch |
|
|
|
250 |
BLR, binary logistic regression; SVM, support vector machine; XGB, extreme gradient boosting; ANN, artificial neural network.
- In figure 2B, the definitions of “Crude”, “Normalizer Scaling”, “MinMax Scaling”, “Standart Scaling” and “Robust Scaling” are missing.
- R) Thank you for the comment. Here, “Crude” refers to the case where the input variables were entered into the raw value, and the rest of them (normalizer, min-max, standard, and robust) were the results when continuous variables were applied to the model using various scaling methods. We added some sentences with relevant references in “Method” section rather than adding this explanation to the footnote of the figure.
“As we described earlier, we aimed to assess the classification performance (HT or no HT) of several ML algorithms using different optimization technique. At first, we randomly divided whole dataset into training and test dataset with 7:3 ratio, of which the proportion of HT in the training and test dataset is maintained similarly in this dataset splitting process. Of input variables, continuous variables were used as the raw values, and categorical variables were encoded using one-hot encoding. In preprocessing process of continuous variables, we used these as raw values (crude method) or using different scaling methods including normalization, min-max scaling, standardization and robust scaling (Figure 1A).20, 21”
- Since their input features include continuous variables as well as categorical variables. How did they scale these different types of variables as input features?
- R) Thank you for the comment. You can be confused because the processing of input variables were not fully described in our initial manuscript. This in an important issue, we have judged this problem to be the same as the one you presented in “issue 1”. We think that Figure 2A has fully explained these problems.
- In Figure 2, why did the authors not implement “Crude”, “Normalizer Scaling”, “MinMax Scaling”, “Standard Scaling” and “Robust Scaling” in BLR, SVM and XGB? Using these scalings of the input variables for all classifiers will make their conclusions more convincing.
- R) Thank you for the comment. As the reviewer’s suggestion, we added some results using additional scaling method for BLR, SVM and XGB algorithms in supplemental result. (Figure S2)
- For a machine learning study, resampling the training and test dataset is essential to reduce sampling bias. A test comparing the four classifiers under multiple repetitions will consolidate their conclusions.
- R) Thank you for the comment. We totally agreed with the reviewer’s opinion. We added the detailed result or the resampling performances of our 4 ML models in the supplemental results (Figure S1).
- The Introduction should be improved:
The two paragraphs in the Introduction are not compact. I think the authors need to present the limitations and challenges of predicting HT with additional references at the end of the first paragraph.
- R) Thank you for the comment. As the reviewer’s suggestion, we added some sentences for the limitation and challenges of predicting HT with additional references.
- The background information about HT and AIS is general. Without sufficient background introduction, it is not easy for readers to intuitively understand why the authors use the baseline characteristics in Table 1.
- R) Thank you for the comment. When we re-read our first manuscript, we were able to reconfirm that, as the reviewer pointed out, the reason for why the ML model was performed with clinical variables was vaguely described. Therefore, we added some sentences for that in the “Introduction” section. The detailed description of this is described in more detail in the response to question 9 below.
- In the second paragraph, the authors need to discuss the shortcomings or limitations of the previous similar studies (such as references [10-14]), and point out the superiority of their model.
- R) Thank you for the comment. As the reviewer’s suggestion, we added some sentences for the importance the limitation of the previous trials as follows.
“Recently, machine learning (ML) or deep learning (DL) algorithms are widely used in medical practice as the name of clinical decision support system. In several studies, usefulness of ML strategy to predict the risk of HT following AIS. Wang et al. reported that neural network model showed the best performance (area under the receiver operating characteristic; AUROC=0.82) to predict symptomatic intracerebral hemorrhage (ICH) following thrombolysis in patients with AIS. In another multicenter trial using Observational Medical Outcomes Partnership Common Data Model, least absolute shrinkage and selection operator regression model showed the AUROC of 0.75 to predict HT. Asadi et al. studied the usefulness of ML algorithms to predict poor outcome in patients with AIS who received endovascular intervention. They suggested that support vector machine (SVM) successfully predict poor outcome, and post-infarct ICH had an important factor for poor prognosis. However, a relatively small amount of study participants (107 subjects) may require additional validation of the study result. Other studies reported high accuracy (~84%) of predicting HT in their stroke cohort, only radiologic markers of MRI were used to perform the ML tasks. In this regard, different ML algorithms have been used to im-prove the prediction of HT after AIS. However, there are no studies showing high predic-tion performance using clinical variables in ML tasks. HT can be divided into symptomatic and asymptomatic cases. Previous studies have reported that not only symptomatic HT, but also asymptomatic HT can affect clinical outcome after AIS. Besides, There are cases where intracranial hemorrhage that occurs after cerebral reperfusion could be asymptomatic,16 and it is difficult to decide when to start antithrombotic treatment in this case.17 Therefore, we limited HT to a radiological definition rather than a clinical definition. We hypothesized that DL algorithms could be better to predict HT after AIS than conventional prediction model. Thus, we aim to assess the important predictor of HT in several ML algorithms, and how to improve the prediction performance of the ML model in this study.”
- Table S1 is interesting and important. It will be useful if the authors can further interpret the biological significance of these top variables in the ML classifiers.
- R) Thank you for the comment. We originally tried to show that variable importance can be identified in the ML task, but we thought that it could reduce the comparison importance of the performance in several ML prediction, which we focused on, and placed it as a supplemental result. However, as the reviewer pointed out, we added this result to the main result, and the related part is described in the "Discussion" section as follows.
“Of our ML models, age, gender, stroke severity, stroke subtype, prior antithrombotics usage, white blood cell count and fasting blood sugar were important variables in predicting HT (Table 3). Age and stroke severity are important prognostic marker for AIS, and also identified as a predictor for HT in other studies. Andrade et al. summarized the important predictors for HT in another clinical trials. Considering that important variables in our ML model are exactly matched with the variables presented in this report, we can suggest that clinically important variables have an important influence on the performance of the classifier even in ML classification.”
- Minor comments:
In Table 1, the “BUN, mg/dL” of the training dataset equaling to “17.7 ± 941” might be a typo.
The abbreviation (for example “AIS” in line 42) should be defined in the first occurrence.
- R) Thank you for the comment. We amended these in the revised manuscript properly.
Round 2
Reviewer 1 Report
All my concerns are solved.